# Vibrational spectroscopy analysis of ligand efficacy in human $M_2$ muscarinic acetylcholine receptor ($M_2R$)

Kota Katayama [1,2,3,8✉], Kohei Suzuki[1,8], Ryoji Suno[4], Ryoji Kise[5], Hirokazu Tsujimoto[6], So Iwata[6], Asuka Inoue [5], Takuya Kobayashi[4,7] & Hideki Kandori [1,2✉]

The intrinsic efficacy of ligand binding to G protein-coupled receptors (GPCRs) reflects the ability of the ligand to differentially activate its receptor to cause a physiological effect. Here we use attenuated total reflection-Fourier transform infrared (ATR-FTIR) spectroscopy to examine the ligand-dependent conformational changes in the human $M_2$ muscarinic acetylcholine receptor ($M_2R$). We show that different ligands affect conformational alteration appearing at the C=O stretch of amide-I band in $M_2R$. Notably, ATR-FTIR signals strongly correlated with G-protein activation levels in cells. Together, we propose that amide-I band serves as an infrared probe to distinguish the ligand efficacy in $M_2R$ and paves the path to rationally design ligands with varied efficacy towards the target GPCR.

[1] Department of Life Science and Applied Chemistry, Nagoya Institute of Technology, Showa-ku, Nagoya 466-8555, Japan. [2] OptoBioTechnology Research Center, Nagoya Institute of Technology, Showa-ku, Nagoya 466-8555, Japan. [3] PRESTO, Japan Science and Technology Agency, 4-1-8 Honcho, Kawaguchi, Saitama 332-0012, Japan. [4] Department of Medical Chemistry, Kansai Medical University, Hirakata 573-1010, Japan. [5] Graduate School of Pharmaceutical Sciences, Tohoku University, Sendai, Miyagi 980-8578, Japan. [6] Department of Cell Biology and Medical Chemistry, Graduate School of Medicine, Kyoto University, Kyoto 606-8501, Japan. [7] Japan Agency for Medical Research and Development, Core Research for Evolutional Science and Technology (AMED-CREST), Chiyoda-ku, Tokyo 100-0004, Japan. [8] These authors contributed equally: Kota Katayama, Kohei Suzuki. ✉email: Katayama.kota@nitech.ac.jp; kandori@nitech.ac.jp

G protein-coupled receptors (GPCRs) are one of the largest family of membrane proteins that induce most of the intracellular biological signalling upon ligand binding[1]. Therefore, understanding the molecular mechanism of GPCR-ligand interaction is vital to elucidating their physiological functions and pathologies. GPCR signalling utilizes a coupling mechanism between the extracellular facing ligand-binding pocket and the cytoplasmic domain of the receptor that selectively interacts with the signalling transducer such as G-proteins, β-arrestins, and various other effectors[2,3]. Furthermore, the different levels of activation of GPCRs are selectively and specifically controlled by the type of ligand, commonly known as efficacy[4–6]. To date, ligands have been categorized into four groups: full agonists, partial agonists, neutral antagonists, and inverse agonists. Understanding the molecular mechanisms that determine the ligand efficacy of GPCRs is important for rational drug design. In addition, discovery of ligands that regulate a target activity has contributed largely to the understanding of both physiological and pathological processes. However, most methods of evaluating ligand efficacy use downstream biochemical and physiological responses that measures the second messenger productivity, protein phosphorylation, and the level of gene expression[7–9], and therefore these methods cannot evaluate the ligand efficacy directly.

Over the last decade, a number of high-resolution X-ray crystal structures of GPCRs have been determined by using lipid cubic phase (LCP). In addition, recent advance of single particle analyses using cryo electron microscopy (cryoEM) provided not only the inactive structures bound with either antagonist or inverse agonist, but also active structures bound with agonists and signal transducers[10–13]. These structures have elucidated key structural changes between the inactive and the active conformations of GPCRs, especially in the extracellular ligand-binding site and the cytoplasmic surface where the effector G-protein interacts[10–13]. The muscarinic acetylcholine receptor 2 ($M_2R$), one of the most extensively studied GPCR has been crystallized with its inverse agonist 3-quinuclidinyl-benzilate (QNB)[14] or N-methylscopolamine (NMS)[15], full agonist Iperoxo (Ixo)[16], and effector $G_o$-protein[17]. Although these studies have provided important insights into the structural changes including the ligand pocket and TM6 movement mediated by the two classes of ligands between inverse agonist and full agonist at the atomic level, its application to a broad variety of ligands with different efficacies, especially partial agonists and neutral antagonists, is extremely challenging. This is partly because efficacy of a ligand is thought to be reflected in changes to conformational equilibria, and thus the presence of multiple states. In addition, these structural methods of X-ray crystallography and cryoEM analysis capture only a snapshot, low-energy conformation, which lack the conformational heterogeneity, therefore these methods cannot fully explain the mechanism of the efficacies.

Spectroscopic techniques such as nuclear magnetic resonance (NMR) and double electron-electron resonance (DEER) have provided insights into the dynamic nature of GPCRs underpinning the conformational plasticity of different efficacy ligand binding[18–23]. Fourier transform infrared (FTIR) spectroscopy has also been successfully applied to examine the structural and functional properties of the photoreceptive GPCR, rhodopsin. Light stimulus-induced difference FTIR spectroscopy combined with low temperature or specific pH values has unveiled various molecular events in the photoactivation processes upon light absorption. The sequential helix movements were monitored by amide-I band corresponding to mostly protein backbone amide carbonyl (C=O) stretching vibration[24,25], local changes in hydrogen bonding were deduced from characteristic $C=O$ stretches of protonated carboxylic acid groups[24,25], and protein

bound water molecules were detected by water O–H/O–D stretching vibrational changes[26]. Furthermore, attenuated total reflection-FTIR (ATR-FTIR) spectroscopy allows to investigate not only light stimulus- but also chemical stimulus-induced protein conformational changes related to their function, such as enzymatic activation[27] or substrate or ligand recognition and binding[28]. In particular, by combining a two-liquid exchange system, perfusion-induced difference ATR-FTIR spectroscopy has been applied to analyze ion-protein and ligand-protein interactions for ion channel and transporter proteins, respectively[29–34]. Another important advantage of this method is that ATR-FTIR generally requires <5 μg of pure protein reconstituted into a lipid bilayer, which makes it highly effective and economical to study GPCRs.

We have recently employed this technique on $M_2R$ to reveal its ligand binding mechanism with its natural agonist, acetylcholine (ACh), and its antagonist, atropine (Atro)[35]. While ACh-bound spectra showed the spectral down-shift in amide-I band (1666 cm$^{-1}$– > 1656 cm$^{-1}$) reflecting to weakening the hydrogen bond between C=O and N–H pairs of peptide backbone, Atro-bound spectra revealed an opposite spectral shift of amide-I band (1643 cm$^{-1}$– > 1656 cm$^{-1}$), which indicates the different conformational changes that occur between an agonist and an antagonist binding to $M_2R$. Furthermore, by tracking the ligand concentration dependence on $M_2R$ activity and ligand binding/dissociation in real time, we could also measure physicochemical properties of ligand binding with $M_2R$. Based on these results, we hypothesize that ATR-FTIR could be positioned as a quick and economical structural analysis tool to examine the ligand binding with GPCRs.

To verify our hypothesis, here we perform systematic ligand binding-induced difference ATR-FTIR spectroscopy measurements on ligands with different efficacies (inverse agonists to full agonists). We observe distinct conformational changes among the agonists, partial agonists, and antagonists in the C=O stetch of amide-I band, which correlates well with G-protein activity in the cells. Time-course ATR-FTIR spectral traces at amide-I band demonstrate differential kinetic patterns: fast dissociation for the full and partial agonists from $M_2R$ and slow or no dissociation for the antagonists and inverse agonists. Together, the amide-I band serves as an infrared probe to distinguish the ligand efficacy of $M_2R$. Additionally, our analysis demonstrate that chemically related ligands exhibit different efficacy.

## Results

**Spectra of the agonists- and partial agonists-bound forms**. We selected four agonists that are structurally similar to ACh, including Metacholine (Meta), Arecholine (Are), Carbamylcholine (Carb), Iperoxo (Ixo), and three partial agonists with no structural similarity to ACh, including Pilocarpin (Pilo), McN-A-343 (McN), and Xanomeline (Xano) (Fig. 1a). In addition, in this study we used the wild-type $M_2R$ fused with thermostabilized apocytochrome b562 (BRIL) at third intercellular loop (ICL3) that has previously been crystallized at resolution of 3.0 Å[15]. All ligand binding-induced ATR-FTIR difference spectra contained noise signals originating from the unbound ligand absorption, distortions from the buffer, absorption changes in water, and the baseline drift due to protein shrinkage (Supplementary Figs. 1–5). After removing these distortions, the baseline-corrected spectra were calculated as shown in Fig. 1b. The ATR-FTIR spectra of the agonist-bound forms were very similar in their spectral features, except for Ixo. As we observed in ACh-bound $M_2R$ spectra, all of the features including three dominant bands 1666 (−)/1656 (+)/1640 (−) cm$^{-1}$ combination bands, positive 1687 cm$^{-1}$ band, and positive 1246 cm$^{-1}$ were detectable in Meta-, Are-, and

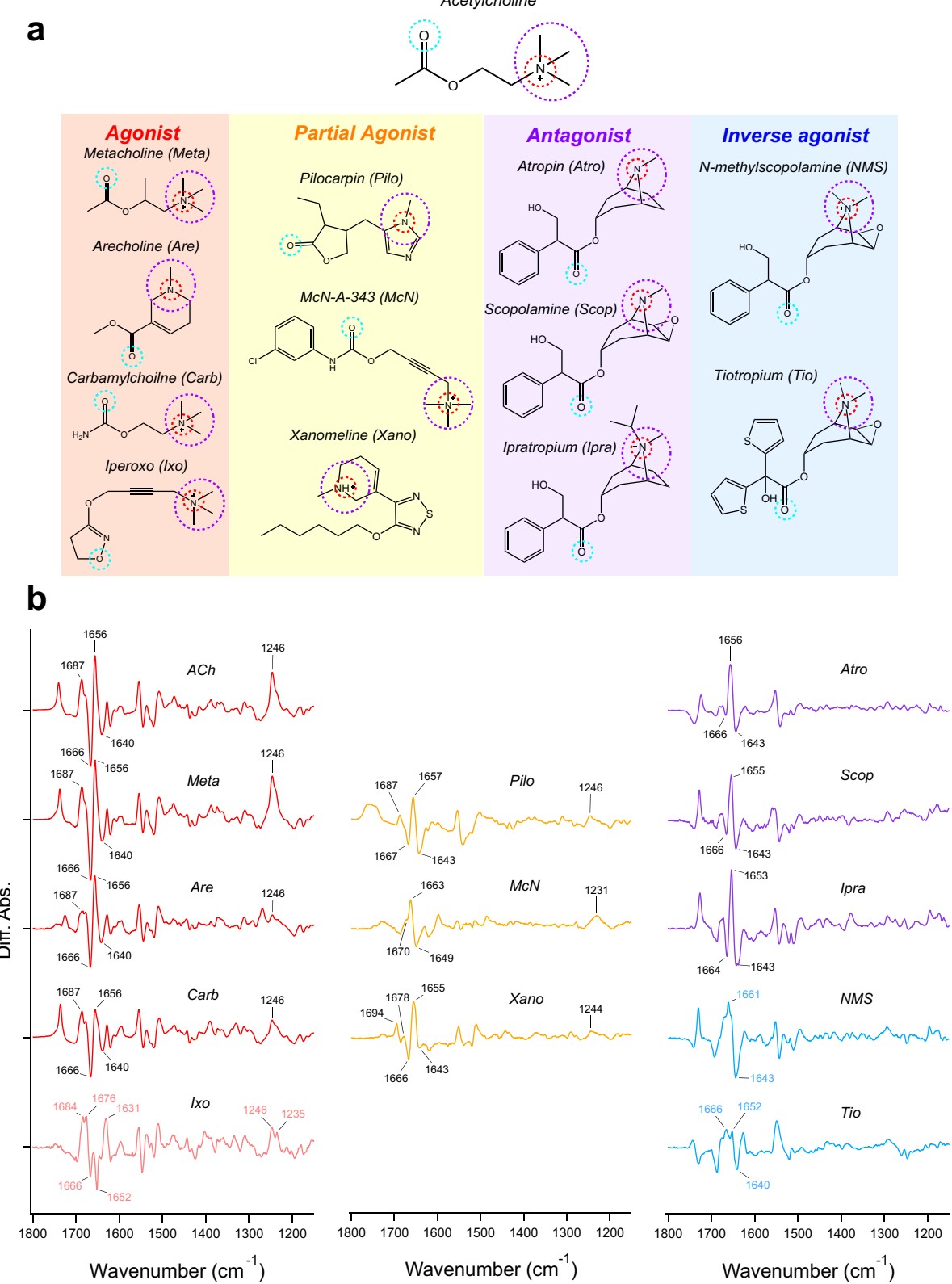

**Fig. 1 Ligand binding-induced difference ATR-FTIR spectra measurement on ligands with different efficacies for M₂R. a** Chemical structures of the ligands used in the ATR-FTIR spectroscopy measurements. Common features among each ligand are marked by dashed circles. **b** Ligand binding-induced difference ATR-FTIR spectra of M₂R bound with various ligands at 293 K. Red, orange, purple, and cyan lines correspond to agonist-, partial agonist-, antagonist-, and inverse agonist-bound spectra measured in H₂O, respectively. Positive and negative bands originate from ligand-bound and ligand-unbound states, respectively. One division of the *y*-axis corresponds to 0.002 absorbance unit.

Carb-bound spectra. A previous study reported that the combination bands ~1650 cm$^{-1}$ are originated from C=O stretch of amide-I band[24,35]. Particularly, a 10 cm$^{-1}$ down-shift from 1666 to 1656 cm$^{-1}$ of amide-I band upon binding of ACh points to an outward movement of TM6 enabling the engagement with G-protein[24,35]. Furthermore, two positive bands at 1687 and 1246 cm$^{-1}$ were tentatively assigned to C=O stretch of Asn404$^{6.52}$ (the numbers in parenthesis denote the residue position in the Ballesteros–Weinstein scheme[36]) side chain and phenolic C–O stretch of tyrosine lid which is comprised of the three conserved tyrosines (Tyr104$^{3.33}$, Tyr403$^{6.51}$, and Tyr426$^{7.39}$), respectively. Both, Asn404$^{6.52}$ and the tyrosine lid constitute the orthosteric ligand binding site of $M_2R$. Since these bands are conserved in Meta-, Are-, and Carb-bound spectra, these agonists exhibit similar binding modes and associated conformational changes in the $M_2R$ protein moiety.

Unlike ACh- and other agonists-bound spectra, Ixo-bound spectra clearly showed a distinctive spectral shift of amide-I band (Fig. 1b). Although the pair bands at 1666 (−)/1652 (−)/1631 (+) cm$^{-1}$ can be attributed to the C=O stretch of amide-I, they showed ~20–30 cm$^{-1}$ spectral downward shift as compared to other agonists-bound spectra. Ixo, being 100-fold more potent than ACh and classified as a super agonist of $M_2R$[37], was used to obtain the active $M_2R$ crystal structure. Previously, solution NMR study in combination with molecular dynamics (MD) simulations revealed large protein conformational changes upon binding of full agonist and super agonist with $M_2R$. Major conformational changes were found in TM5 and TM6, which would be influenced by slight changes in the orthosteric-binding site of $M_2R$[22]. Namely, Ixo binds deeper into the ligand binding pocket of $M_2R$ than Ach via a hydrogen bond with Asn404$^{6.52}$ and a water molecule near Asn404$^{6.52}$ [22]. Interestingly, the Ixo-bound spectra shows a positive 1684 cm$^{-1}$ band which tentatively originates from the C=O stretch of Asn404$^{6.52}$ with a 3 cm$^{-1}$ downward shift as compared to ACh-bound spectra (Fig. 1b). This suggests a stronger hydrogen bond interaction of Ixo with Asn404$^{6.52}$ than ACh. The observed changes in the hydrogen bond strength particular to Ixo-binding is likely the reason of the distinct TM conformational change detected as 20–30 cm$^{-1}$ downward shift of amide-I band.

In contrast to agonists, partial agonists-bound spectra display relatively different spectral features. At ~1650 cm$^{-1}$ region, the band intensity and the population of the combination band of amide-I (1666 (−)/1656 (+)/1640 (−) cm$^{-1}$ in ACh-bound spectra) were altered upon partial agonists binding. However, the band frequencies were nearly identical to that of agonists-bound spectra. In Pilo- and McN-bound spectra, while the bands at 1667 (−) or 1670 (−) cm$^{-1}$ corresponding to the apo form are decreased, 1643 (−) or 1649 (−) cm$^{-1}$ bands intensities were enhanced. These pattern of spectral shift of amide-I bands are similar to that of antagonist Atro-bound spectra (Fig. 1b). In contrast, partial agonist Xano-bound spectra had 1666 (−)/1655 (+)/1643 (−) cm$^{-1}$ combination band intensity similar to that of agonists-bound spectra. The decrease in band intensity of amide-I caused by partial agonists was consistent with their lower efficacy towards $M_2R$. Thus, the spectral shift pattern of the amide-I band suggests that it is reflected in the conformational equilibrium between the inactive state and active states of $M_2R$ as indicated from previous FTIR study of rhodopsin[24,25].

While similar protein structural changes were observed between agonists and partial agonists which result in equilibrium shift from inactive to active states, partial agonists-dependent conformational changes around ligand binding site were also observed. Similar to ACh-bound spectra, Pilo-bound spectra shows two positive bands at 1687 and 1246 cm$^{-1}$, which originates from Asn404$^{6.52}$ and the tyrosine lid, respectively.

Unlike Pilo-bound spectra, McN-bound spectra does not show the positive band at 1687 cm$^{-1}$, and the positive band at 1246 cm$^{-1}$ shifts to 1231 cm$^{-1}$. For Xano bound spectra we also observed a 7 cm$^{-1}$ up shift in Asn404$^{6.52}$ signal to 1694 cm$^{-1}$ and a 2 cm$^{-1}$ down-shift in tyrosine lid signal to 1244 cm$^{-1}$. These results suggest a different hydrogen bond strength between the nitrogen of Asn404$^{6.32}$ and the acetyl oxygen (in Pilo and McN) or sulphur (in Xano) (Fig. 1a). This is one of the key reasons for the deferential activation of $M_2R$ by various classes of ligands.

**Spectra of the antagonist- and inverse agonist-bound forms.** Next, we investigated the conformational changes induced by antagonists and inverse agonists. Previous FTIR study showed that Atro-bound spectra clearly exhibited the different spectral shift pattern of amide-I as compared to the ACh-bound spectra. The two positive bands originating from Asn404$^{6.52}$ and the tyrosine lid were absent, which might suggest a weaker interaction of Atro with Asn404$^{6.52}$ and a loose connecting triad of tyrosine lid[35]. With respect to spectral features, Scopolamine (Scop)- and Ipratropium (Ipra)-bound spectra were similar with Atro-bound spectra (Fig. 2b purple curves). Each antagonist-bound spectra possesses the combination bands of amide-I at 1666 (−)/1655 (+)/1643 (−) cm$^{-1}$ for Scop and 1664 (−)/1653 (+)/1643 (−) cm$^{-1}$ for Ipra. As expected, the two ligand-binding site specific positive bands at 1687 and 1246 cm$^{-1}$ are missing in both spectra, strongly indicating that all three antagonists (Atro, Scop, and Ipra) bind to the orthosteric site of $M_2R$ and induce a similar conformational change in the TM region. These results are consistent with the structural similarity between these antagonists (Fig. 1a). These antagonists differ only at cationic amine group, which forms an electrostatic interaction with Asp103$^{3.32}$ [38].

In contrast to antagonists, inverse agonists-bound spectra show completely different spectral features as compared to both antagonists- and agonists-bound spectra (Fig. 1b cyan curves). For N-methylscopolamine (NMS)-bound spectra, dominant peaks at 1661 (+)/1643 (−) cm$^{-1}$ will correspond to amide-I band at 1656 (+)/1643 (−) cm$^{-1}$ as observed in antagonists-bound spectra. However, the corresponding negative 1666 cm$^{-1}$ band is lacking in NMS-bound spectra. Additionally, the band ~1660 cm$^{-1}$ is broadened. For tiotropium (Tio)-bound spectra, in addition to the amide-I pair bands at 1652 (+)/1640 (−) cm$^{-1}$, a new positive band was observed at 1666 cm$^{-1}$. The observed distinct spectral changes of amide-I band indicate conformational heterogeneity in both NMS- and Tio-bound structures of $M_2R$. This is consistent with previous NMR studies that revealed two conformations of $M_2R$ upon binding with Tio[22]. Notably, both inverse agonists-bound spectra showed no positive bands at 1687 and 1246 cm$^{-1}$ originating from Asn404$^{6.52}$ and tyrosine lid, which is consistent with antagonists-bound spectra. Taken together, different patterns of spectral shift of amide-I and spectral changes of functional group of amino acids of the orthosteric-binding site of $M_2R$ were observed, depending on the efficacy of the bound ligand.

**Correlation between relative intensities of the amide-I bands and the activation of G protein.** To quantitatively examine the change in the amide-I band depending on the ligand efficacy, the ratio of the band strength (1656 (+)/1666 (−) cm$^{-1}$ in case of ACh-bound spectra, active state component) at high frequency to the band strength at low frequency (1656 (+)/1640 (−) cm$^{-1}$ in case of ACh-bound spectra, inactive state component) was calculated by equation in Fig. 2a. The ratio of the amide-I band was assumed to be the ligand efficacy (Fig. 2b, c). Strikingly, all agonists have an amide-I band ratio >1. The amide-I band ratio

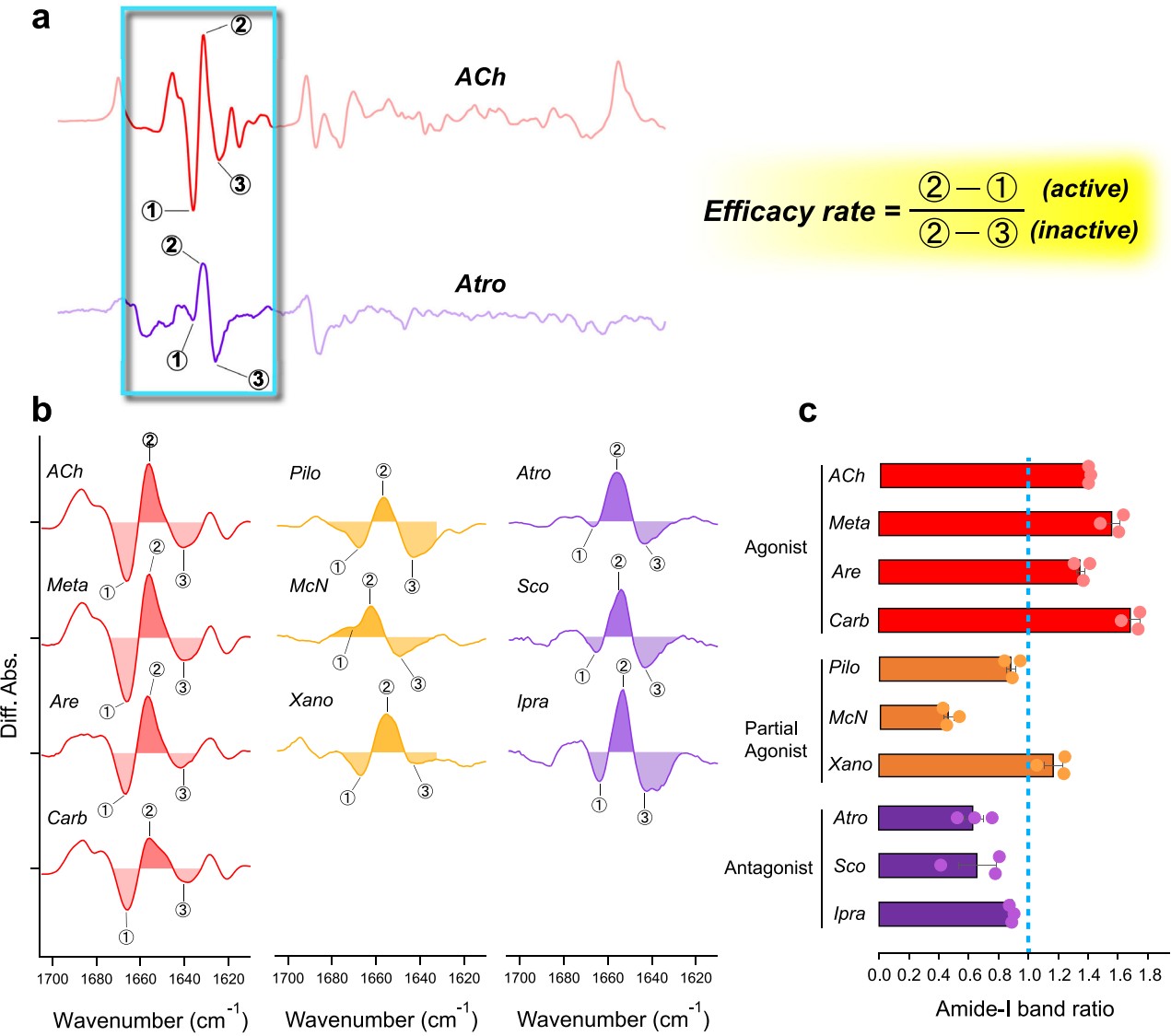

**Fig. 2 Ligand-dependent spectral changes in the α-helical region of M$_2$R and relative populations of active and inactive states of M$_2$R. a** Ligand binding-induced difference ATR-FTIR spectra in the 1800–1200 cm$^{-1}$ region, especially focusing on amide-I band region, which are taken from Fig. 1B. Red and purple lines correspond to ACh- and Atro-bound spectra, respectively. To quantitatively examine the change in the amide-I band specific to ligand efficacy, the ratio between the band strength of (2)1656 (+)/(1)1666 (−) cm$^{-1}$ of ACh-bound spectra at high frequency and (2)1656 (+)/(3)1640 (−) cm$^{-1}$ of ACh-bound spectra at low frequency is calculated and reported as efficacy rate. **b** Ligand-dependent spectral changes of amide-I band originating from C=O stretch of α-helix in 1700–1620 cm$^{-1}$ region. Red, orange, and purple lines correspond to agonist-, partial agonist-, and antagonist-bound spectra, respectively. One division of the y-axis corresponds to 0.002 absorbance unit. **c** Relative populations of active and inactive states upon ligand binding in M$_2$R derived by the equation of efficacy rate from **a**. The error values were calculated from three replicate experiments. Cyan dotted line indicates the value = 1.

for both the partial agonists and the antagonists were <1, with the exception of Xano. The ligands, Ixo (super agonist) and NMS and Tio (inverse agonists) which gave a complex spectral variation, were excluded from the present analysis. Compared to Pilo and McN among partial agonists, Xano stabilizes a higher population of active-like M$_2$R conformation, which is characterized by the outward movement of TM6. On the other hand, among the antagonists, Ipra has a higher ratio of active conformation than Atro and Scop.

We assumed that the amide-I band ratio correlates with M$_2$R signalling efficacy. To quantitatively compare the two parameters, we calculated changes in the intensity of the amide-I band by infrared spectroscopy and measured the efficacies of each ligand toward G$_i$-protein activation using a NanoBiT G-protein dissociation assay (Supplementary Fig. 6)[39]. The functional assays

show that three agonists; Meta, Are, and Carb which have similar chemical structures, show almost identical G$_i$-protein activity as ACh-bound M$_2$R (Fig. 3a). Ixo represents higher G-protein activation than ACh, which is consistent with the reported property of super agonist (Fig. 3a)[37]. We found that partial agonists, Pilo, McN, and Xano exhibit decreased G$_i$ signalling relative to ACh. By contrast, the tested antagonists showed poor (Ipra; 6.3% of ACh) or undetectable (Atro and Scop) G$_i$-dissociation activity (Fig. 3a).

Next, we plotted the amide-I band ratio for agonists, partial agonists, and antagonists against their relative G$_i$-protein efficacy to ACh. The amide-I band ratio correlated well with agonist efficacy in promoting G$_i$ coupling (E$_{max}$) (Fig. 3b), but did not correlate with their potency values, pEC50 (Supplementary Fig. 7). It should be noted that while the antagonist showed a minimal or

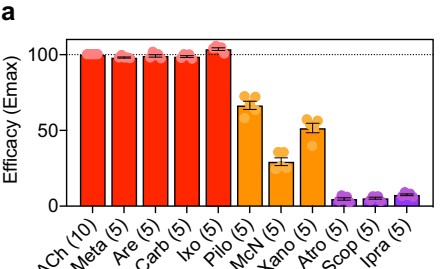
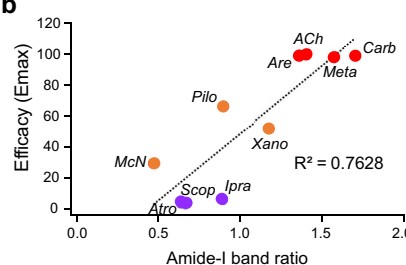

**Fig. 3 Correlation between ligand efficacy and amide-I band ratio. a** Efficacies of the different $M_2R$ ligands toward $G_i$ activation in the NanoBiT G-protein dissociation assay. $E_{max}$ values were calculated from the concentration-response sigmoidal curves in Extended Data Fig. 6 and were normalized to that of ACh performed in parallel. Bars and error bars represent mean and SEM, respectively, of 5 or 10 (shown in parenthesis) independent experiments with each performed in duplicate. **b** Correlation between ligand efficacy and the relative intensities of the amide-I bands. The relative intensities of the amide-I bands are calculated by the equation of efficacy rate, which is derived from Fig. 2c. Ligand efficacy is determined by the NanoBiT G-protein dissociation assay from **a**. Agonists (ACh, Meta, Are, and Carb) are shown as red circles, partial agonists (Pilo, McN, and Xano) as orange circles, and antagonists (Atro, Scop, and Ipra) as purple circles.

no apparent effect on $G_i$-protein activation, the amide-I band ratio unexpectedly did not go below 0.5. This means that the antagonists-bound spectra contain a certain percentage of active-like state in $M_2R$ due to conformational heterogeneity as indicated from previous NMR study[22] though primarily inactive state is presented. Nevertheless, the band shift changes in amide-I[24,25] can be used as an infrared probe of agonist efficacy that promotes $G_i$-coupling.

Structural changes are also reflected in the amide-II band of the protein backbone, a coupled of the N–H in-plane bending and C–N stretching vibrations[40]. The ratio of the band intensity was calculated in the same way as for amide-I band (Supplementary Fig. 8a, b). Unlike amide-I band, no remarkable difference in the intensity ratio of the amide-II band was observed for all ligands. In fact, the amide-II percent population did not correlate with agonist efficacy in promoting $G_i$ coupling (Supplementary Fig. 8c). We considered that it was difficult to discuss the pure amide-II band because the region ~1550 cm$^{-1}$ where the amide-II band appeared was overlapped by the vibrational bands of the amino acid functional groups and the ligand itself (Supplementary Fig. 1).

On the other hand, in addition to the changes in the protein backbone reflected in the amide-I band, ligand-dependent spectral changes were also observed in the phenolic C–O stretch of tyrosine lid. As already described in Fig. 1b, the positive 1246 cm$^{-1}$ band was observed in all agonist- and partial agonist-bound spectra except for McN, but not in antagonist-bound spectra (Supplementary Fig. 9). We recently compared difference FTIR spectra between unlabelled and 2-$^{13}$C labelled ACh, and assigned the band at 1246 cm$^{-1}$ as the mixed band of C–O stretch of ACh and possibly C-O stretch of tyrosine lid[41]. Thus, the C–O stretch of tyrosine lid can be also used as an infrared probe of the ligand efficacy.

**Implications of ligand-dependent dissociation kinetics from $M_2R$.** Additional insights on the discrimination of ligand efficacy from ATR-FTIR measurements can be obtained by examining the ligand dissociation kinetics ($k_{off}$) from $M_2R$ (Fig. 4a). In a previous study, the time evolution of the difference ATR-FTIR spectra over the course of the experiment showed different dissociation events between ACh and Atro with $M_2R$[35]. While the band intensity of amide-I in ACh-bound spectra decreased gradually after exchanging the buffer without ACh, the band intensity in Atro-bound spectra did not decrease during the dissociation phase artificially caused by the buffer exchange. These behaviors are consistent with their $K_i$ values (ACh; 10 µM[42], Atro; 0.8 nM[43,44]). As shown in Fig. 4b, dissociation kinetics of Meta and Carb was similar to ACh-bound $M_2R$. These

results are consistent with similar ligand-bound spectral features observed as shown in Fig. 1b. Although Are-bound $M_2R$ also displayed similar spectral features to that of ACh-bound spectra (Fig. 1b), its dissociation rate suggests slower kinetics (Supplementary Fig. 11). This is probably due to the steric hindrance caused by the Are-specific chemical structure possessing tetrahydropyridine, which makes it difficult to dissociate from the receptor. On the other hand, Ixo exhibited the slowest dissociation kinetics. This result also indicates difficulty in dissociating from the receptor and corresponds to previous NMR results suggesting that Ixo binds more deeply in the ligand binding pocket than ACh[22].

For partial agonists, while we observe fast dissociation kinetics of Pilo and McN like for ACh, Xano exhibited extremely slow $k_{off}$ (Supplementary Fig. 11). This can be likely explained by the tetrahydropyridine ring present in the chemical structures of both Are and Xano[45]. In addition, Xano is known to act as the strongest G-protein biased agonist of $M_2R$[46], and one of the underlying reasons for its strong G-protein biased signalling could be a longer dissociation kinetics of the ligand-receptor complex. With the exception of Are, Ixo, and Xano, all agonists and partial agonists we used in the present study dissociated from $M_2R$ by buffer exchange, whereas all antagonists and inverse agonists did not show any dissociation from the receptor, which are consistent with their strong binding (Supplementary Fig. 11). Given that the spectra of the inverse agonists-bound form are completely unique in shape compared to other ligands-bound spectra, ligand binding-induced difference ATR-FTIR spectroscopy appears to be a versatile tool to distinguish between antagonist and inverse agonist binding to GPCRs.

**Discussion**

Here, taking advantage of ATR-FTIR spectroscopy, we investigated the conformational changes of $M_2R$ when bound to various ligands. The amide-I band shift pattern of α-helical C=O stretch demonstrated that different classes of $M_2R$ ligands altered the population between inactive and active states, with the respect to the protein conformational changes, including an outward movement of TM6[24,25]. Furthermore, vibrational signals originating from functional group of key amino acids (Asn404[6.52] and especially tyrosine lid (Tyr104[3.33], Tyr403[6.51], and Tyr426[7.39])) that constitute the ligand binding pocket were varied by different ligand efficacy (Fig. 5a, b).

In the previous NMR studies of $^{13}$C$^{ε}$H$_3$-methionine-labeled β$_2$AR[18], turkey β$_1$AR[19], µOR[20], and α$_{1A}$-AR[21], the chemical shifts

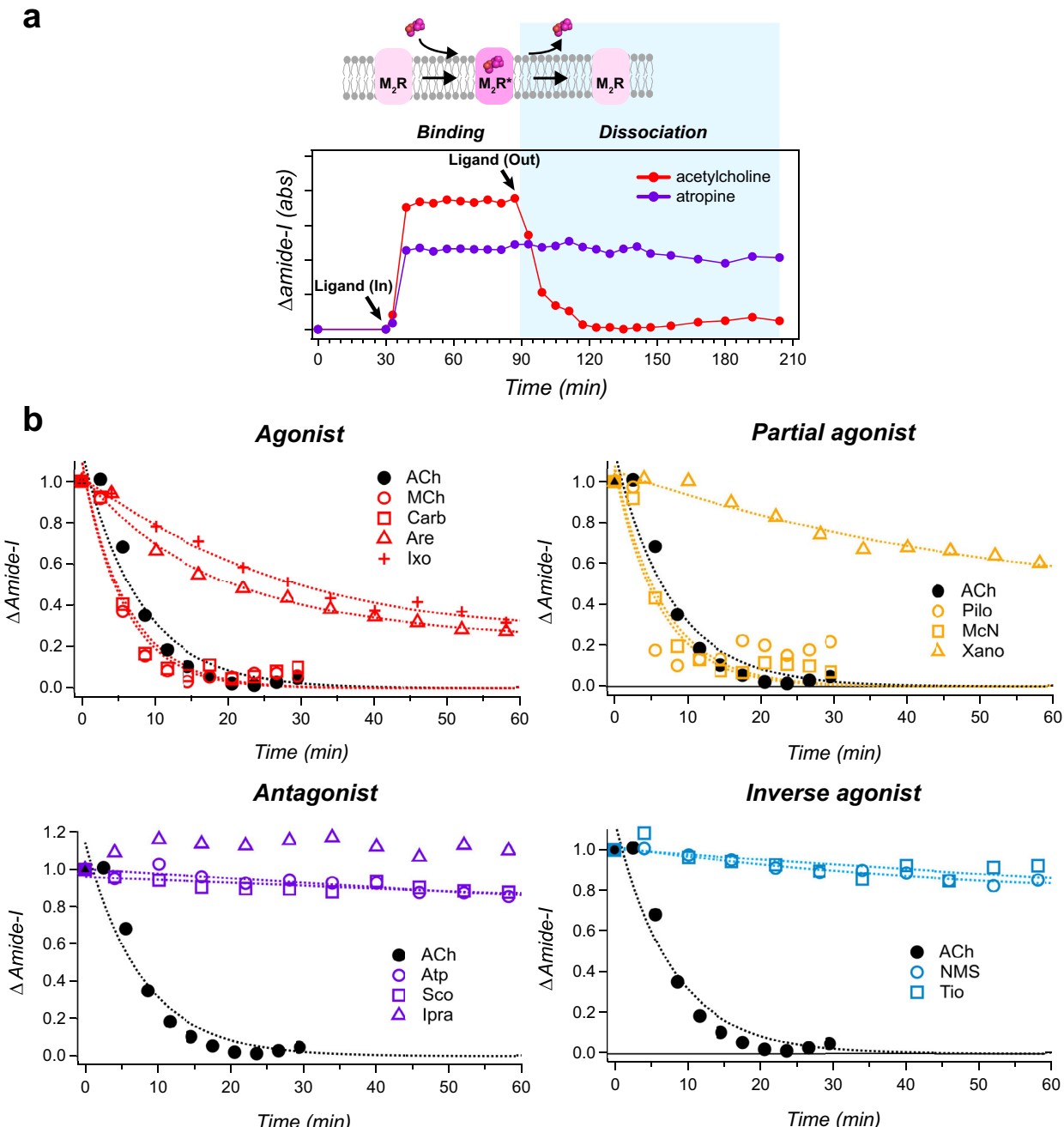

**Fig. 4 Ligand-dependent dissociation kinetics on M₂R. a** Time trace of the integrated absorbance signal in the amide-I band (red circle, ACh; purple circle, Atro) taken from Katayama et al.[35]. Each ligand (1 mM concentration) is added at 30 min time point and is washed out at 90 min time point by exchanging to the buffer without ligand at a flow rate of 0.6 mL min⁻¹ through a flow cell, of which the temperature is maintained at 20 °C by circulating water. The arrows in the illustration mark the time points of ligand-In and ligand-out via buffer exchange, respectively. Ligand dissociation phase is highlighted by light blue. **b** Time trace of the integrated absorbance signal at dissociation phase in the amide-I band for each ligand. Time-dependent difference ATR-FTIR spectra upon ligand dissociation in the amide-I region (1680–1630 cm⁻¹) are shown in Supplementary Fig. 10. Black red, orange, purple, and cyan correspond to ACh, agonist, partial agonist, antagonist, and inverse agonist, respectively. Dotted lines represent the fitting curve obtained by single exponential function.

reflecting receptor conformations for different ligands showed a linear correlation with their ligand efficacies. On the other hand, a similar NMR analysis for M₂R did not show a strong correlation with ligand efficacy, suggesting that M₂R showed a complex conformational heterogeneity[22]. In the present study, by combining ATR-FTIR spectroscopy with functional cell-based Gᵢ-protein assays, we found a correlation between the amide-I percent population and ligand efficacy for both full and partial agonists, but some

ligands such as Ixo or Xano showed complex spectral features and were outliers. Furthermore, all the amide-I percent population of antagonists-bound spectra did not show linear correlation with cell-based Gᵢ-protein assay. Rather, the amide-I percent population analysis indicates that the equilibrium proportions of active and inactive states of M₂R is similar between partial agonists and antagonists. On the other hand, Asn404⁶·⁵² and tyrosine lid are likely involved in not only direct ligand binding, but also allosteric

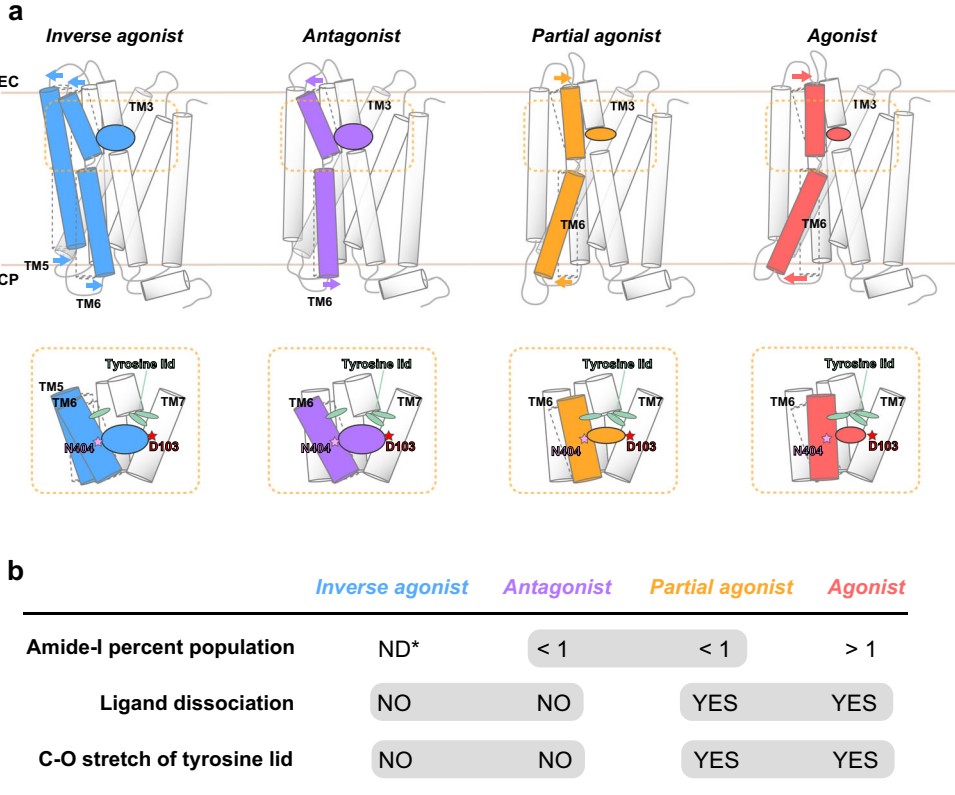

**Fig. 5 Proposed conformational changes in $M_2R$ upon binding of ligands with different efficacies. a** (Upper) Schematic of $M_2R$ TM6 and TM5 conformational states upon binding of orthosteric ligands with different efficacies. TM6 features an open conformation in the extracellular side and a closed conformation in the cytoplasmic side upon binding with inverse agonist and antagonist. In contrast, TM6 features a closed conformation in the extracellular side and an open conformation in the cytoplasmic side upon binding with both agonist and partial agonist, which causes the opposite movement with both inverse agonist and antagonist. Yellow dotted line indicates the orthosteric ligand binding pocket. (Lower) Schematic of $M_2R$ ligand binding pocket surrounded by TM3, 5, 6, and 7. Two key amino acids (Asn404[6.52] and Asp103[3.32]) and tyrosine lid are depicted by star and oval markers, respectively. Binding of either inverse agonist or antagonist opens the extracellular region of TM6, resulting in loosening of the tyrosine lid, whereas both agonist- and partial agonist-bound forms compacts the ligand-binding pocket, which induces the formation of tyrosine lid that excludes solvent entry. **b** Summary of the differences in the ATR-FTIR analysis of $M_2R$ with various ligand efficacies. By comprehensive comparing the three factors (amide-I band ratio, ligand dissociation, and C–O stretch of tyrosine lid) extracted from ATR-FTIR spectroscopy analysis, different ligand efficacy can be distinguished.

activation of G-protein because the changes in Asn404[6.52] and tyrosine lid are observed in the agonists- and partial agonists-bound spectra, but not in the antagonists-bound spectra. These results demonstrate that $M_2R$ shows conformational plasticity, and therefore future application of same ATR-FTIR measurements to other class A type of GPCRs such as $\beta_2AR$[18], turkey $\beta_1AR$[19], $\mu OR$[20], and $\alpha_{1A}$-AR[21] may lead to a confirmation.

Another noteworthy aspect of the present study is that the inverse agonists-bound spectra for both NMS and Tio are completely different from other ligands-bound spectra, especially in the amide-I band region. The result of NMS-bound spectra showed a broadening of the amide-I band, whereas the Tio-bound spectra was a bilobed amide-I band, indicative of unique conformational changes as compared with other types of ligands. So, what are the inverse agonist-specific structural changes compared to an antagonist, even though both ligands reduce the activation of GPCRs? Recently determined structure of $M_1R$ bound with Atro[47] clearly showed large conformational differences as compared to Tio-bound $M_1R$[48] at the extracellular end of TM5, where a slight inward displacement at TM5 was observed in the Atro-bound form relative to Tio binding. Most likely, the two arene ring of Tio causes steric clashes with TM5, which prevents the inward movement of TM5 at the end of extracellular side like in Atro-bound $M_1R$. Thus, one of the two positive bands of

amide-I at 1666 cm$^{-1}$ in the Tio-bound spectra may correspond to a change in TM5 at extracellular region, while the other band at 1652 cm$^{-1}$ being specific to TM6 motion.

On the other hand, NMS does not have two arene rings like Tio, but rather a very similar chemical structure to Scop (Fig. 1a). Nonetheless, how NMS can exhibit efficacy as an inverse agonist? The only difference is the presence or absence of a methyl group in tropane alkaloid between Scop and NMS. So, does the difference in efficacy depend solely on the presence or absence of methyl groups between Scop (antagonist) and NMS (inverse agonist)? To investigate this possibility, we measured ligand-binding induced difference ATR-FTIR spectroscopy of Oxitropium (Oxitro) and N-butylscopolamine (NBS), possessing an ethyl or a butyl group in tropane alkaloid, respectively. NBS-bound spectra was similar to Scop-bound spectra, while Oxitro-bound spectra exhibited a complex (broadening and/or bilobed) spectral feature like in NMS-bound spectra, especially at the amide-I band (Fig. 6a). This result suggests that NBS acts as an antagonist and Oxitro as an inverse agonist. The inactive structures of Atro-bound form of $M_1R$[47] and NMS-bound form of $M_2R$[15] show a common involvement of Asp103[3.32] in the interaction with the tropane alkaloid (Fig. 6b). Thus, to function as an inverse agonist, the length of the tropane alkaloid side chain should not be too long or too short, and only methyl or ethyl groups adopts an energetically favorable

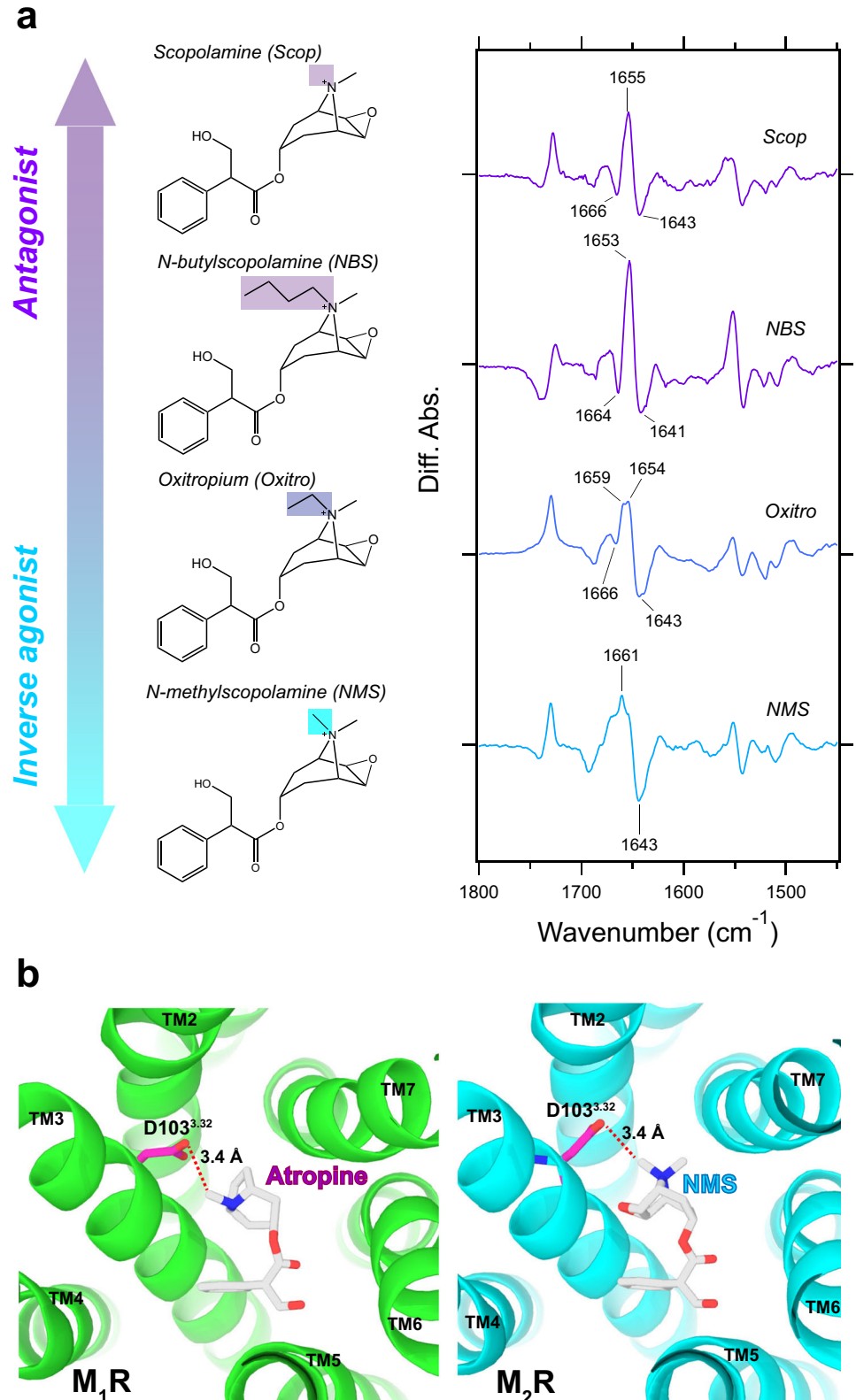

**Fig. 6 Comparison between the scopolamine derivatives. a** Chemical structures of the scopolamine derivative ligands (Scopolamine (Scop), *N*-butylscopolamine (NBS), Oxitropium (Oxitro), and *N*-methylscopolamine (NMS)) used in the current FTIR spectroscopic studies, and each ligand bound spectra in the 1800–1450 cm$^{-1}$ region. The group of quaternary ammonium derivative positions are highlighted. Ligand binding-induced difference ATR-FTIR spectra are measured in H$_2$O at 293 K. Positive and negative bands correspond to ligand-bound and ligand-unbound states, respectively. One division of the y-axis corresponds to 0.0025 absorbance unit. **b** Comparison between Atro-bound structure in M$_1$R (green, PDB: 6WJC)[47] and NMS-bound structure in M$_2$R (cyan, PDB: 5YC8)[15] at the view from extracellular side. The amino acid residues and ligands are depicted by sticks, and TM helices are depicted by ribbons. Hydrogen bond between Asp103[3.32] and amine group of both ligands are shown by red dotted lines with hydrogen bond length as labels.

conformation to connect tightly with Asp103$^{3.32}$, resulting in reducing the receptor activity (Supplementary Fig. 12).

Despite recent developments in structural biology methods, distinguishing the protein conformational changes specific to the binding of an inverse agonist or antagonist remains difficult. This is partly because the crystal structures of both inverse agonist- and antagonist-bound forms show common structural changes, including an outward movement of TM6 at the extracellular side and a corresponding inward shift of the intracellular end of TM6, as seen in several structures of class A GPCR-ligand complexes[49]. On the other hand, the ATR-FTIR spectroscopic analysis performed in this study clearly distinguishes inverse agonists- and antagonists-bound forms of $M_2R$ by detecting the difference in the spectral feature of amide-I band caused by the difference in carbon chain length, suggestive of its broad implications in rational drug design of GPCRs (Supplementary Fig. 12).

Although we succeeded in distinguishing orthosteric ligand efficacy in $M_2R$, all of them that we used in this study are water-soluble ligands. Our measurement system, which uses lipid reconstituted samples, is incompatible with measurement for lipid-soluble ligands because of the effect of protein shrinkage associated with lipid–ligand interaction. Therefore, future improvements in measurement system is clearly needed to evaluate the efficacy of any ligand. Moreover, since our method proved to be able to evaluate the efficacy of ligands for only one type of receptor, it is necessary to demonstrate the general applicability of our method to a wide range of other GPCRs in the future.

In summary, we have described a new method for the quantitative evaluation of efficacy of four types ligands including agonist, partial agonist, antagonist, and inverse agonist from the vibrational perspective of amide-I band change in $M_2R$ embedded into lipid environment using ATR-FTIR spectroscopy. This biophysical technique can also provide the physicochemical parameters such as dissociation constant and dissociation rate constant of ligand simultaneously. Overall, the vibrational spectroscopic method reported herein provides a promising strategy for measuring ligand efficacy at a wide variety of GPCRs.

## Methods

**Protein expression and purification.** The wild-type $M_2R$ fused with BRIL at ICL3 position ($M_2R$) was expressed and purified as described previously[15], except for some minor modifications for reconstitution into the membrane. Briefly, C-terminally His-tagged $M_2R$-BRIL with the hemagglutinin (HA) signal sequence followed by an N-terminal FLAG tag was expressed in Sf9 insect cells. Cells were infected at a density of $3–4 \times 10^6$ cells/mL and grown for 48 h at 27 °C. Sf9 cells were lysed by osmotic shock in the presence of 10 μM atropine (Sigma-Aldrich). The lysed membranes were solubilized using a buffer containing 30 mM HEPES-NaOH (pH 7.5), 0.75 M NaCl, 5 mM imidazole, 1% (w/v) n-dodecyl-β-D-maltopyranoside (DDM; anatrace), 0.2% sodium cholate (Wako), 1 mg mL$^{-1}$ iodoacetamide (Dojindo), and Complete Protease inhibitor (Roche) for 1 h at 4 °C. The supernatant was isolated by ultracentrifugation for 30 min at 140,000×$g$, and incubated with Ni-NTA Sepharose Superflow resin (Qiagen) overnight at 4 °C. After binding, the resin was washed with Ni-NTA wash buffer: 30 mM HEPES-NaOH (pH 7.5), 0.75 M NaCl, 0.1% (w/v) DDM, 0.02% (w/v) sodium cholate, 5 mM imidazole, and 10 μM atropin. The protein was then eluted with Ni-NTA elution buffer: 30 mM HEPES-NaOH (pH 7.5), 0.75 M NaCl, 0.1% (w/v) DDM, 0.02% (w/v) sodium cholate, 5 mM imidazole, 10 μM atropine, 500 mM imidazole. The eluate was supplemented with 2 mM calcium chloride and loaded onto an anti-FLAG M1 affinity resin (Sigma). The receptor was eluted from the anti-FLAG M1 affinity resin with a buffer of 20 mM HEPES-NaOH (pH 7.5), 0.1 M NaCl, 0.01% (w/v) DDM, 10 μM atropine, 0.2 mg mL$^{-1}$ FLAG peptide, and 5 mM EDTA. Finally, protein was purified by Superdex 200 Increase size exclusion column (GE Healthcare) in a buffer of 20 mM HEPES-NaOH (pH 7.5), 0.1 M NaCl, 0.01% (w/v) DDM.

**Protein reconstitution.** For ATR-FTIR measurements, detergent-solubilized $M_2R$ was reconstituted into asolectin (Sigma) liposomes with a 20-fold molar excess. The detergent molecule was removed by incubation with Bio-beads SM-2 (Bio-Rad, CA, USA). After removal of Biobeads, the lipid-reconstituted $M_2R$ was collected by ultracentrifugation for 30 min at 222,000×$g$ at 4 °C. After several cycles of wash/spin, lipid-reconstituted $M_2R$ was suspended in a buffer composed of 5 mM phosphate (pH 7.5) and 10 mM KCl.

**Measurement of ligand binding-induced difference ATR-FTIR spectroscopy.** A 2 μL aliquot of the lipid-reconstituted $M_2R$ suspensions was placed on the surface of a silicon ATR crystal (three internal total reflection, Smith Detection, UK. After it was dried in a gently natural drying, the sample was rehydrated with a solvent containing 200 mM phosphate (pH 7.5) buffer with 140 mM NaCl, 3 mM MgCl$_2$ at a flow rate of 0.6 mL min$^{-1}$ through a flow cell, of which the temperature was maintained at 20 °C by circulating water. ATR-FTIR spectra were first recorded at 2 cm$^{-1}$ resolution, using an FTIR spectrometer (Bio-rad FTS7000, Agilent, CA, USA) equipped with a liquid nitrogen-cooled MCT detector (an average of 768 interferograms). After the FTIR spectrum had been recorded in the second buffer with 1 mM ligand, the difference FTIR spectrum was calculated by subtracting the data obtained for the first and second buffer. The cycling procedure was repeated two to seven times, and the difference spectra were calculated as the average of the presence minus absence spectra of ligand. The spectral contributions of the unbound ligand, the protein/lipid shrinkage, and water/buffer components were corrected (Supplementary Figure 2–5).

**NanoBiT G-protein dissociation assay.** $M_2R$-induced G-protein dissociation was measured by a NanoBiT-G-protein dissociation assay[39], in which the interaction between a Gα subunit and a Gβγ subunit was monitored by the NanoBiT system (Promega). Specifically, a NanoBiT-$G_{i1}$ protein consisting of $Gα_{i1}$ subunit fused with a large fragment (LgBiT) at the α-helical domain and an N-terminally small fragment (SmBiT)-fused $Gγ_2$ subunit with a C68S mutation was expressed along with untagged $Gβ_1$ subunit and $M_2R$. HEK293A cells were seeded in a 10-cm culture dish at a concentration of $2 \times 10^5$ cells mL$^{-1}$ (10 mL per well in DMEM (Nissui) supplemented with 10% fetal bovine serum (Gibco), glutamine, penicillin, and streptomycin), 1 day before transfection. Transfection solution was prepared by combining 25 μL (per dish hereafter) of polyethylenimine (PEI) Max solution (1 mg mL$^{-1}$; Polysciences), 1 mL of Opti-MEM (Thermo Fisher Scientific), and a plasmid mixture consisting of 1 μg $M_2R$ (or an empty plasmid for mock transfection), 500 ng LgBiT-containing $Gα_{i1}$ subunit, 2.5 μg $Gβ_1$ subunit, and 2.5 μg SmBiT-fused $Gγ_2$ subunit (C68S). After an incubation for 1 day, the transfected cells were harvested with 0.5 mM EDTA-containing Dulbecco's PBS, centrifuged and suspended in 10 mL of HBSS containing 0.01% bovine serum albumin (BSA; fatty acid–free grade; SERVA) and 5 mM HEPES (pH 7.4) (assay buffer). The cell suspension was dispensed in a white 96-well plate at a volume of 80 μL per well and loaded with 20 μL of 50 μM coelenterazine (Carbosynth) diluted in the assay buffer. After a 2 h incubation at room temperature, the plate was measured for baseline luminescence (Spectramax L, Molecular Devices) and a titrated test ligand (20 μL; 6X of final concentrations) was manually added. The plate was immediately read at room temperature for the following 5 min as a kinetics mode, at measurement intervals of 20 s. The luminescence counts over 3–5 min after ligand addition were averaged and normalized to the initial count. The fold-change values were further normalized to that of vehicle-treated samples and used to plot the G-protein dissociation response. Using the Prism 8 software (GraphPad Prism), the G-protein dissociation signals were fitted to a four-parameter sigmoidal concentration-response curve. For each replicate experiment, the parameter Span (= Top–Bottom) of individual ligands was normalized to ACh and the resulting $E_{max}$ values were used as efficacy.

**Statistics and reproducibility.** All functional and statistical data were analyzed using GraphPad Prism v.9.0 (Graphpad Software) and shown as mean ± s.e.m. from at least three independent experiments. Dissociation kinetics curves were evaluated with a single exponential function.

**Reporting summary.** Further information on research design is available in the Nature Research Reporting Summary linked to this article.

## Data availability

The data supporting the findings of this study are available in the article, Supplementary Information, and if applicable, from the corresponding author on request. In addition, all the data supporting the findings of this manuscript have been deposited in Figshare.com (https://doi.org/10.6084/m9.figshare.16608511 or https://figshare.com/s/a16d3e63d43ba9be9c74)[50].

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

## Acknowledgements

We appreciate Drs. S. Gulati and M. Singh for helpful comments on the manuscript. We thank Kayo Sato, Shigeko Nakano, and Ayumi Inoue (Tohoku University) for their assistance with plasmid preparation, maintenance of cell culture and cell-based GPCR assays. This work was supported by JSPS KAKENHI, Japan, grant numbers 18K14662 and 21H01883 to K.K., 18H03986 and 19H04959 to H.K., 15K08268, 19H03428 and 21H05112 to R.S., 19H05777 to S.I., and 21H05113 to A.I.; the Naito Science & Engineering Foundation to K.K.; the Takahashi Industrial and Economic Research Foundation to K.K.; the Takeda Science Foundation to T.K. and R.S.; the Naito Foundation to T.K.; the Koyanagi Foundation to T.K.; the Basis for Supporting Innovative Drug Discovery and Life Science Research (BINDS) from the Japan Agency for Medical Research and Development (AMED) under grant number JP20am0101079 to S.I. and JP20am0101095 to A.I.; AMED under grant number JP20gm0910007, JP20am0401020, and JP20ak0101103 to T.K. and the LEAP JP20gm0010004 to A.I.; and Japan Science and Technology Agency (JST) PRESTO (JPMJPR19G4) to K.K.

## Author contributions

K.K., K.S., R.S., A.I. and H.K. contributed to the study design. R.S. and H.T. expressed samples in Sf9 and purified them. K.K. and K.S. reconstituted samples for spectroscopic measurements. K.K. and K.S. conducted ATR-FTIR spectroscopic measurements. R.K. and A.I. performed NanoBiT G-protein dissociation assay. K.K. prepared the initial manuscript and K.K., K.S., R.S., R.K., S.I., A.I., T.K. and H.K. wrote the paper with input from all the authors. All authors discussed and commented on the manuscript.

## Competing interests

The authors declare no competing interests.

**Additional information**

