## [Transparent Peer Review File · Communications Biology]

Reviewers' comments:

Reviewer #1 (Remarks to the Author):

The manuscript uses infrared techniques to monitor conformational change within the muscarinic receptor (M2R) with respect to the efficacy of ligands. Generally, there is a clear trend, although the diversity of the chemistry of each group of ligands is rather limited. I did find the use of IR to understand dissociation constants (Figure 4) quite interesting and complements this work and other findings. Another strength of the paper was the low amounts of protein used and it is incorporated in a liposome (bilayer). Overall the flow of data and discussion is well done. There are a number of errors that made the paper a little difficult to appreciate. These can be easily corrected. I will list my criticisms.

Possible major corrections:

Line 65 I do not agree that the strength of affinity will result in a low-energy stable conformation; depends on many factors and I think the perception of a low energy state is more a reduction in heterogeneity. The following sentence is hinting that the methods of xray and possibly cryoEM select a single (low-energy) state and miss the diversity, so the fact that a low energy state or single state is selected by some methods is a criticism worth mentioning. The lines near here need correcting.

Line 85 I do not practice this technique, and I think for the reader, clarification of what the "stretch of the amide-I band" means – eg do the frequencies relate to weakening or strengthening of interactions (hydrogen bonds)? It would help give context.

Line 223 Is it correct to plot potency for the antagonists? Very tiny differences compared to vehicle (extended fig 6). Further the error bars are small or rather uniform for efficacy and potency of the antagonists, indeed all ligands. Are these analyses correct? This also applies to the incomplete titrations eg McN. I cannot find statistics statements.

Line 225 to 227 "These results indicate that M2R shows more conformational plasticity than other class A type of GPCRs such as β 2AR18, turkey β 1AR19, μ OR20, and α 1A-AR21" Essentially I agree that the M2R has a greater diversity (as pointed out in discussion and partly arising from the NMR publication (ref 22)), but I am not sure you make this conclusion based on comparing this infrared work here to the other receptors which did not use the same method.

Minor corrections.

Line 62 not clear what the two classes of ligand are.

Line 65 their strong affinities (not high their binding) (also cause not causes)

Line 86 opposite (not oppose)

Line 107 I do not think the word "combined" is needed

Line 114-116 you need to cite the previous study

Line 119-121 I am not clear on what the signals really are. Is the 1687 cm^{-1} for Asn404 the side-chain or main-chain carbonyl; is the 1246 cm^{-1} C-O stretch of tyrosine the aromatic C-OH group or is this a typo and should be (mainchain) C=O. So please make very clear.

Line 146 instead of negative 1643 or 1649 write 1643 (-) or 1649 (-) as you have been.

Line 147 and 148 write "antagonist Atro-bound....partial agonist Xano-bound" (makes it easier to follow)

Line 159 "For xano-bound spectra we also observed a 7 cm^{-1}" I think this line is referring to xano-bound, but it is not stated?

Line 178 refer to figure 1

Line 202 to 204. Isn't Xano in the wrong spot in the text. It is >1 ? "with the exception of Xano" should be shifted to the next line "...and the antagonist were <1 , with the exception of Xano"

Lin 223 correlate not correlated

Line 307 You cannot say "exactly identical" ...similar.

Line 310 While not critical, I gather you did not test Oxitro to show it is an inverse agonist? If there is data or literature include.

Figure 1 and 6. I think the structure of N-methylscopolamine is correct but the way the oxygen is bonded to the tropane is incorrect for scopolamine, NBS and Oxitro.

Reviewer #2 (Remarks to the Author):

Katayama et al. present a report on the use of attenuated total FTIR spectroscopy to study the interactions of ligands with the M2 muscarinic receptor, a class A human GPCR. This work builds on an earlier paper by Katayama and colleagues published in J Phys Chem Lett in 2019. In the earlier 2019 publication, the methodology for the FTIR experiments was introduced and spectra of M2 were reported with two agonist complexes and an antagonist complex. The submitted manuscript expands on that paper by presenting FTIR data of M2 in complexes with a wider range of ligands, including additional agonists, partial agonists, antagonists, and inverse agonists. The manuscript also presents additional measurements of the dissociation of ligands as viewed by FTIR spectroscopy.

The presented data and method are interesting and the expanded data set with a wider range of ligand efficacies compared with the 2019 publication does, in principle, allow for a more systematic exploration of the effects of ligand efficacies and chemical structures on the FTIR spectra and may justify a follow up paper to the 2019 publication. The data look promising and are rich with potential information. The authors demonstrate that using FTIR data could become a convenient approach to measure a "fingerprint" of changes in GPCR structure upon complex formation with different drugs. However, there are several significant concerns about the analysis and presentation that make me hesitant to support publication of this work without major revisions.

The core of this paper is summarized by Figures 2 and 3. Here the authors are attempting to correlate ligand efficacy with an analysis of the fine structure of FTIR spectra for the 10 different complexes studied. The authors attempt to quantify the differences in the amide-I region by calculating an "Efficacy Rate" parameter. While this parameter does appear to show clear and consistent differences between agonists and all other ligands, this analysis fails to distinguish between partial and full agonists or partial agonists, antagonists, and inverse agonists. The calculated "Efficacy Rate" shows no difference between partial agonists and antagonists (Fig. 2b). Thus the analysis shows that while this data can identify agonists, it cannot further distinguish between different ligand efficacies. I understand the authors' approach to using information from the amide-I region, but the analysis does not show a clear linear correlation with drug efficacy. This information is possibly contained in the spectra, and I wonder if including additional features of the spectra outside of the amide-I region in this analysis would more closely relate to ligand efficacy. Without a clearer correlation, use of this

analysis is limited to distinguishing between agonists and all other ligands. Either a different analysis should be provided that is better correlated with ligand efficacies or these figures should be revised to show that one can distinguish agonists but that no clear linear correlation appears to exist that spans the entire range of drug efficacies.

In Figure 3b, it seems that the equation used to calculate the linear fit to the experimental data did not include data with antagonists. If one were to take the residuals of the data points, all the antagonists would fall well below the line rather than being distributed above and below the line, indicating either a systematic error in the experiments or a systematic error in the data analysis. This plot should be removed as it does not support the authors' attempts to show a relationship between drug efficacy and analysis of the amide-I region.

On the bottom of page 9 and top of page 10, the authors write "In contrast, from the amide-I percent population, the antagonists can induce an equilibrium shift to the active state at a certain rate, but has a minimal or no apparent effect on Gi-protein activation . . ." This statement does not make sense and is not consistent with known pharmacology or other biophysical measurements. This statement should be removed.

In the introduction, the authors state, "Spectroscopic techniques such as nuclear magnetic resonance (NMR) and double electron-electron resonance (DEER) . . . however both spectroscopy techniques require . . .and/or site-directed mutations . . ." This is a disingenuous statement given that the authors employed a heavily modified GPCR for their own studies, a GPCR with a non-native protein fused into the third intracellular loop. This statement is also nearly word-for-word identical to what was written in the 2019 J Phys Chem Lett paper. I realize that this is a general introduction, but the text is nearly identical to what was written before, which is likely a violation of the journal policies. Given this and given that the authors are themselves using a receptor that is heavily modified compared with the native protein, the authors need to remove this statement from their introduction.

The authors should also state in the results section the exact identify of the employed construct. This information is in the methods section but also needs to be stated up front in the results.

Related to this, in the methods section the authors cite a 2018 Nature Chem Bio paper that employed the same protein and purification strategy. However, the receptor in the 2018 study also included an S110R stabilizing point mutation. If this was used in the current study it needs to be stated explicitly both in the results and methods sections. The 2018 Nature Chem Bio paper also states that the S110R mutation increases the affinities for antagonists. This would certainly impact the presented data, especially the data in Figure 4. Therefore this absolutely needs to be clarified in the text, otherwise no one will be able to reproduce these data if they are using a receptor without the S110R mutation.

The Figure 2 x-axis states "amide 1 percent population" but this does not make sense with the units. The left and right-most units are 0.0 and 2.0, which would then be 0% to 2%, which is not what I believe the authors intended.

Regarding the data in Figure 4, it should be explained more clearly how these experiments are done. I'm assuming that at 30 minutes ligand is added to solution, but this is not explicitly described in the paper or made clear in the figure, which could be shown by indicating this time point with an arrow. Other important experimental details are also not described for these experiments, such as the flow rate, the amount of ligand added, when ligand was added. Also related to Figure 4, the authors should report what effective off rates were calculated based on this data and it would be interesting to compare those number with any available literature data from surface plasmon resonance or other techniques, if it is publicly available (if not, then it would be acceptable to not include this comparison).

Regarding the model presented in Figure 5, unless assignments of the fine structure have been

confirmed, this model is not presently supported by the data. In the 2019 J Phys Chem Lett paper, the authors tentatively assigned several lines to specific amino acids or groups of amino acids, such as the tyrosine lid. In the present work, the authors assume these assignments are correct, even though there were stated as being tentative assignments only in the 2019 paper. The spectra are complex and there are many structural changes that occur in GPCRs between antagonist-bound and agonist-bound conformations, including changes for the same amino acid types that have been tentatively assigned but in regions of the protein that are different than what the authors claim. Without more definitive assignment information, showing this model makes too many assumptions and therefore it should be removed.

Reviewer #3 (Remarks to the Author):

The paper reports a vibrational spectroscopic study using ATR-FTIR of the human M2 muscarinic acetylcholine receptor (M2R). The work involves an extensive comparison of agonists, antagonists, and inverse agonists, in which it is shown that they C=O stretching amide-I band is correlated with the ligand binding efficiency. The results are explained in terms of shifting of the conformational equilibrium between active and inactive states of the M2R receptor. As the major finding, it is proposed that agonists and partial agonists forward shift the equilibrium to the active state, and conversely for the antagonists and inverse agonists. All of the results are plausible, interesting, and significant to the field of GPCR biology and pharmacology.

Generally the paper is well-written with some minor language anomalies. The figures are clear and illustrate the main points of the research. The strength of the paper is the comparison of agonists, antagonists, and inverse agonists and the connection of the downstream activity in terms of an activation equilibrium. In the future one would like to see a more detailed comparison of the ATR-FTIR bands in terms of the spectra of the active and in active states. The main weakness is overlooking previous work with rhodopsin as a prototype for Family A GPCRs, which has likewise been interpreted in terms of shifting of the activation equilibrium by conversion of the ligand (retinal) from an inverse agonist to an agonist, which is sensitive to the membrane environment. The previous work of Vogel et al. for rhodopsin clearly shows the shifting of the conformational equilibrium upon the light activation, as manifested in the difference FTIR spectral shifts as well as the UV visible difference spectra. Please see the following papers:

Mahalingam, M., Martínez-Mayorga, K., Brown, M. F., Vogel, R. (2008), Two Protonation Switches Control Rhodopsin Activation in Membranes, Proc. Natl. Acad. Sci. U.S.A. 105 17795-17800. <https://doi.org/10.1073/pnas.0804541105>

Zaitseva, E., Brown, M. F., and Vogel, R. (2010), Sequential Rearrangement of Interhelical Networks Upon Rhodopsin Activation in Membranes: The Meta IIa Conformational Substate, J. Am. Chem. Soc. 132, 4815-4821. <https://doi.org/10.1021/ja910317a>

The paper should make reference to the earlier rhodopsin work of Vogel at all at several junctures in the text as indicated below.

Major comments:

Page 4, second paragraph - please refer to the previous findings of Vogel et al. for rhodopsin here.

Pag 5, second paragraph - with regard to the amide-I band please refer to Vogel et al. papers.

Page 7, end of first paragraph - here it is essential to refer to the above-mentioned Vogel et al. papers, which show how the rhodopsin activation equilibrium is shifted upon conversion of retinal from

an inverse agonist to an agonist.

Page 10, top paragraph - please refer to above mentioned Vogel et al. papers.

Page 11 beginning of discussion - please refer to Vogel et al.

With these revisions the paper should be ready for publication in Nature Communications Biology.

Replies to the Comment of Reviewer 1

We are thankful to you for your valuable comments and suggestions, which are very helpful to us in improving the manuscript.

We reply to your comments as follows.

Major comments

[Comment 1]

Line 65 I do not agree that the strength of affinity will result in a low-energy stable conformation; depends on many factors and I think the perception of a low energy state is more a reduction in heterogeneity. The following sentence is hinting that the methods of xray and possibly cryoEM select a single (low-energy) state and miss the diversity, so the fact that a low energy state or single state is selected by some methods is a criticism worth mentioning. The lines near here need correcting.

[Reply]

Thank you for your useful comment. We revised the manuscript as follows.

This is partly because of their low affinities to GPCR receptors resulting in conformational heterogeneity. ~~most of the ligands have similar chemical and structural properties and high their binding affinities ($K_i \leq 10$ nM) causes the GPCR receptors to adapt their low energy stable conformations.~~ In addition, these structural techniques methods of X-ray crystallography and cryoEM analysis capture only a snapshot, low-energy conformation, which lack the conformational heterogeneity, therefore these methods cannot fully explain the mechanism of the efficacies.

[Comment 2]

Line 85 I do not practice this technique, and I think for the reader, clarification of what the “stretch of the amide-I band” means – eg do the frequencies relate to weakening or strengthening of interactions (hydrogen bonds)? It would help give context.

[Reply]

We revised the manuscript as follows, thank you.

While ACh-bound spectra showed the spectral down-shift in amide-I band (1666 cm^{-1} -- $> 1656\text{ cm}^{-1}$) reflecting to weakening the hydrogen-bond between C=O and N-H pairs of

peptide backbone, ~~large spectral changes of the amide I band (the C=O stretch of the α -helix)~~, Atro-bound spectra revealed an opposite spectral shift of amide-I band ($1643\text{ cm}^{-1} \rightarrow 1656\text{ cm}^{-1}$), which indicates the different conformational changes that occur between an agonist and an antagonist binding to M_2R .

[Comment 3]

Line 223 Is it correct to plot potency for the antagonists? Very tiny differences compared to vehicle (extended fig 6). Further the error bars are small or rather uniform for efficacy and potency of the antagonists, indeed all ligands. Are these analyses correct? This also applies to the incomplete titrations eg McN. I cannot find statistics statements.

[Reply]

Extended Data Fig. 6 & 7 show correct data. Although apparent error bars look similar, each value slightly differs. In terms of incomplete titrations such as McN, We could not take more concentration points due to the stock solution concentration of McN, but the present fitting curve shows sigmoidal fitting and reproducibility of E_{max} , so we believe that the measurement is sufficient.

[Comment 4]

Line 225 to 227 “These results indicate that M_2R shows more conformational plasticity than other class A type of GPCRs such as β_2AR^{18} , turkey β_1AR^{19} , μOR^{20} , and $\alpha_{1A}AR^{21}$ “ Essentially I agree that the M_2R has a greater diversity (as pointed out in discussion and partly arising from the NMR publication (ref 22)), but I am not sure you make this conclusion based on comparing this infrared work here to the other receptors which did not use the same method.

[Reply]

Thank you for your useful comment. According to the suggestion, we revised the manuscript as follows, thank you.

These results indicate that M_2R shows more conformational plasticity ~~than other class A type of GPCRs such as β_2AR^{18} , turkey β_1AR^{19} , μOR^{20} , and $\alpha_{1A}AR^{21}$~~ as indicated from previous NMR study²².

Minor comments

[Comment 1]

Line 62 not clear what the two classes of ligand are.

[Reply]

Thank you for your useful comment. According to the suggestion, we revised the manuscript as follows, thank you.

Although these studies have provided important insights into the structural changes including the ligand pocket and TM6 movement mediated by the two classes of ligands between inverse agonist and full agonist at the atomic level, its application to a broad variety of ligands with different efficacies, especially partial agonists and neutral antagonists, is extremely challenging.

[Comment 2]

Line 65 their strong affinities (not high their binding) (also cause not causes)

[Reply]

We revised the manuscript as follows, thank you.

This is partly because of their low affinities to GPCR receptors resulting in conformational heterogeneity. ~~most of the ligands have similar chemical and structural properties and high their binding affinities ($K_i \leq 10$ nM) causes the GPCR receptors to adapt their low energy stable conformations.~~ In addition, these structural techniques methods of X-ray crystallography and cryoEM analysis capture only a snapshot, low-energy conformation, which lack the conformational heterogeneity, therefore these methods cannot fully explain the mechanism of the efficacies.

[Comment 3]

Line 86 opposite (not oppose)

[Reply]

We revised the manuscript as follows, thank you.

While ACh-bound spectra showed the spectral down-shift in amide-I band (1666 cm^{-1} -- > 1656 cm^{-1}) reflecting to weakening the hydrogen-bond between C=O and N-H pairs of

peptide backbone, ~~large spectral changes of the amide I band (the C=O stretch of the α -helix)~~, ATR-bound spectra revealed an opposite spectral shift of amide-I band ($1643\text{ cm}^{-1} \rightarrow 1656\text{ cm}^{-1}$), which indicates the different conformational changes that occur between an agonist and an antagonist binding to M_2R .

[Comment 4]

Line 107 I do not think the word “combined” is needed

[Reply]

We revised the manuscript as follows, thank you.

All ligand binding-induced ATR-FTIR difference spectra contained ~~combined~~ noise signals originating from the unbound ligand absorption,

[Comment 5]

Line 114-116 you need to cite the previous study

[Reply]

We added the reference as follows, thank you.

A previous study reported that the combination bands around 1650 cm^{-1} are originated from C=O stretch of amide-I band^{24,35}. Particularly, a 10 cm^{-1} down-shift from 1666 to 1656 cm^{-1} of amide-I band upon binding of ACh points to an outward movement of TM6 enabling the engagement with G-protein^{24,35}.

24: Mahalingam, M., Martínez-Mayorga, K., Brown, M. & Vogel, R. Two protonation switches control rhodopsin activation in membranes. *Proc. Natl. Acad. Sci. U. S. A.* **105**, 17795-17800 (2008).

35: Katayama, K., Suzuki, K., Suno, R., Tsujimoto, H., Iwata, S., Kobayashi, T. & Kandori, H. Ligand binding-induced structural changes in the M_2 muscarinic acetylcholine receptor revealed by vibrational spectroscopy. *J. Phys. Chem. Lett.* **10**, 7270-7276 (2019).

[Comment 6]

Line 119-121 I am not clear on what the signals really are. Is the 1687 cm^{-1} for Asn404

the side-chain or main-chain carbonyl; is the 1246 cm⁻¹ C-O stretch of tyrosine the aromatic C-OH group or is this a typo and should be (mainchain) C=O. So please make very clear.

[Reply]

We revised the manuscript as follows, thank you.

two positive bands at 1687 and 1246 cm⁻¹ were tentatively assigned to C=O stretch of Asn404^{6.52} (the numbers in parenthesis denote the residue position in the Ballesteros-Weinstein scheme³⁶³) side chain and phenolic C-O stretch of tyrosine lid which is comprised of the three conserved tyrosines (Tyr104^{3.33}, Tyr403^{6.51}, and Tyr426^{7.39}), respectively.

[Comment 7]

Line 146 instead of negative 1643 or 1649 write 1643 (-) or 1649 (-) as you have been.

[Reply]

We revised the manuscript as follows, thank you.

while the bands at 1667 (-) or 1670 (-) cm⁻¹ corresponding to the apo form are decreased, ~~negative~~ 1643 (-) or 1649 (-) cm⁻¹ bands intensities were enhanced.

[Comment 8]

Line 147 and 148 write “antagonist Atro-bound....partial agonist Xano-bound” (makes it easier to follow)

[Reply]

We revised the manuscript as follows, thank you.

These pattern of spectral shift of amide-I bands are similar to that of antagonist Atro-bound spectra (Fig. 1b). In contrast, partial agonist Xano-bound spectra had 1666 (-)/1655 (+)/1643 (-) cm⁻¹ combination band intensity similar to that of agonists-bound spectra.

[Comment 9]

Line 159 “For xano-bound spectra we also observed a 7 cm⁻¹....” I think this line is referring to xano-bound, but it is not stated?

[Reply]

We revised the manuscript as follows, thank you.

For Xano bound spectra wWe also observed a 7 cm⁻¹ up shift in Asn404^{6.52} signal to 1694 cm⁻¹ and a 2 cm⁻¹ down-shift in tyrosine lid signal to 1244 cm⁻¹.

[Comment 10]

Line 178 refer to figure 1

[Reply]

We revised the manuscript as follows, thank you.

These results are consistent with the structural similarity between these antagonists (Fig. 1a).

[Comment 11]

Line 202 to 204. Isn't Xano in the wrong spot in the text. It is >1? "with the exception of Xano" should shifted to the next line "...and the antagonist were <1, with the exception of Xano"

[Reply]

We revised the manuscript as follows, thank you.

The ~~amide-I percent population~~ ratio of the amide-I band was assumed to be the ligand efficacy (Fig. 2b and 2c). Strikingly, all agonists have an amide-I ~~percent population~~ band ratio >1, ~~with the exception of Xano~~. The amide-I ~~percent population~~ band ratio for both the partial agonists and the antagonists were <1, with the exception of Xano.:

[Comment 12]

Line 223 correlate not correlated

[Reply]

We revised the manuscript as follows, thank you.

but did not correlated with their potency values,

[Comment 13]

Line 307 You cannot say “exactly identical” ...similar.

[Reply]

We revised the manuscript as follows, thank you.

NBS-bound spectra was similar~~exactly identical~~ to Scop-bound spectra,

[Comment 14]

Line 310 While not critical, I gather you did not test Oxitro to show it is an inverse agonist? If there is data or literature include.

[Reply]

As you point out, we have not measured cell-based activity against Oxitropium (Oxitro) and N-butylscopolamine (NBS). This is mainly because inverse agonist activity (=GPCR constitutive activity) cannot be detected with the NanoBiT G-protein assay used in this study. In addition, even in the TGF α -shedding assay (Inoue, A., Ishiguro, J., Ktamura, H., Arima, N., Okutani, M., Shuto, A., Higashiyama, S., Ohwada, T., Arai, H., Makide, K. & Aoki, J. TGF α shedding assay: an accurate and versatile method for detecting GPCR activation. *Nature Methods* **9**, 1021-1029 (2012)), which can detect GPCR constitutive activity, the preliminary data (data not shown) indicate that the signal window is small, and evaluation is difficult. Therefore, we could not draw any conclusions in this study, but only make suggestions.

[Comment 15]

Figure 1 and 6. I think the structure of N-methylscopolamine is correct but the way the oxygen is bonded to the tropane is incorrect for scopolamine, NBS and Oxitro.

[Reply]

We changed the figures, thank you.

Replies to the Comment of Reviewer 2

We are thankful to you for your critical reading of our manuscript. We reply to your comments as follows.

Major comments

[Comment 1]

The core of this paper is summarized by Figures 2 and 3. Here the authors are attempting to correlate ligand efficacy with an analysis of the fine structure of FTIR spectra for the 10 different complexes studied. The authors attempt to quantify the differences in the amide-I region by calculating an “Efficacy Rate” parameter. While this parameter does appear to show clear and consistent differences between agonists and all other ligands, this analysis fails to distinguish between partial and full agonists or partial agonists, antagonists, and inverse agonists. The calculated “Efficacy Rate” shows no difference between partial agonists and antagonists (Fig. 2b). Thus, the analysis shows that while this data can identify agonists, it cannot further distinguish between different ligand efficacies. I understand the authors’ approach to using information from the amide-I region, but the analysis does not show a clear linear correlation with drug efficacy.

[Reply]

We appreciate the suggestion. Certainly, Fig. 2 & 3 are important appealing points that this paper claims, but equally, the difference in dissociation dynamics of ligands from receptor shown in Fig. 4 is also an important factor to distinguish the ligand efficacy. As pointed out, it is difficult to distinguish between partial agonist and antagonist from the amide-I band ratio. On the other hand, while partial agonists exhibit dissociation from the receptor (with except for Xano), antagonists show a significantly slower or no dissociation manner. Thus, the difference in the dissociation rate between them is also an important factor to distinguish the ligand efficacy.

[Comment 2]

I wonder if including additional features of the spectra outside of the amide-I region in this analysis would more closely relate to ligand efficacy. Without a clearer correlation, use of this analysis is limited to distinguishing between agonists and all other ligands.

[Reply]

Thank you for your useful comment. According to the suggestion, we have performed

the same analysis for the amide-II band (a coupled of the N-H in-plane bending and C-N stretching vibrations), which reflect changes in the protein backbone as well. We then plotted the correlation between the ratio of the band intensities and the agonist efficacy in promoting G_i coupling (Extended Data Fig. 8 (revised manuscript)). Unexpectedly, unlike the amide-I band, the amide-II band did not correlate with the G-protein activation efficacy, which we consider is due to the fact that the vibrational frequency region where the amide-II band appears is covered by the infrared absorption bands of other amino acid functional groups and the ligand itself.

In addition, we focused on a band that we consider to be the phenolic C-O stretch of tyrosine lid as a band that shows ligand-dependent spectral changes (Extended Data Fig. 9 (revised manuscript)). In other words, the presence or absence of the 1246 cm^{-1} band indicates the difference in ligand efficacy.

[Comment 3]

In Figure 3b, it seems that the equation used to calculate the linear fit to the experimental data did not include data with antagonists. If one were to take the residuals of the data points, all the antagonists would fall well below the line rather than being distributed above and below the line, indicating either a systematic error in the experiments or a systematic error in the data analysis. This plot should be removed as it does not support the authors' attempts to show a relationship between drug efficacy and analysis of the amide-I region.

[Reply]

We appreciate the suggestion and carried out the linear fit including antagonists (Fig. 3b (revised manuscript)).

[Comment 4]

On the bottom of page 9 and top of page 10, the authors write “In contrast, from the amide-I percent population, the antagonists can induce an equilibrium shift to the active state at a certain rate, but has a minimal or no apparent effect on G_i -protein activation . . .” This statement does not make sense and is not consistent with known pharmacology or other biophysical measurements. This statement should be removed.

[Reply]

We apologize for the confusion and corrected as follows.

It should be noted that while the antagonist showed a minimal or no apparent effect on G_i -protein activation, the amide-I band ratio unexpectedly did not go below 0.5. This suggests that the antagonists-bound spectra contain a certain percentage of active-like state in M_2R due to ~~In contrast, from the amide I percent population, the antagonists can induce an equilibrium shift to the active state at a certain rate, but has a minimal or no apparent effect on G_i protein activation (Fig. 3b, purple).~~ These results indicate that M_2R shows ~~more~~ conformational heterogeneity—~~than other class A type of GPCRs such as β_2AR ¹⁸, turkey β_1AR ¹⁹, μOR ²⁰, and $\alpha_{1A}AR$ ²¹~~ as indicated from previous NMR study²² though primarily inactive state is presented.

[Comment 5]

In the introduction, the authors state, “Spectroscopic techniques such as nuclear magnetic resonance (NMR) and double electron-electron resonance (DEER) . . . however both spectroscopy techniques require . . .and/or site-directed mutations . . .” This is a disingenuous statement given that the authors employed a heavily modified GPCR for their own studies, a GPCR with a non-native protein fused into the third intracellular loop. This statement is also nearly word-for-word identical to what was written in the 2019 J Phys Chem Lett paper. I realize that this is a general introduction, but the text is nearly identical to what was written before, which is likely a violation of the journal policies. Given this and given that the authors are themselves using a receptor that is heavily modified compared with the native protein, the authors need to remove this statement from their introduction.

[Reply]

We revised the manuscript as follows, thank you.

Spectroscopic techniques such as nuclear magnetic resonance (NMR) and double electron-electron resonance (DEER) have provided insights into the dynamic nature of GPCRs underpinning the conformational plasticity of different efficacy ligand binding¹⁸⁻²³. Fourier transform infrared (FTIR) spectroscopy has also been successfully applied to examine the structural and functional properties of the photoreceptive GPCR, rhodopsin. Light stimulus-induced difference FTIR spectroscopy combined with low temperature or specific pH values has unveiled various molecular events in the photoactivation processes upon light absorption. The sequential helix movements were monitored by amide-I band corresponding to mostly protein backbone amide carbonyl (C=O) stretching vibration^{24,25}, local changes in hydrogen bonding were deduced from characteristic C=O stretches of

protonated carboxylic acid groups^{24,25}, and protein bound water molecules were detected by water O-H/O-D stretching vibrational changes²⁶ (Furutani Y. et al. (2003)). However, ~~both spectroscopy techniques require large quantities of pure protein, isotopic labelling and/or site directed mutations, which might not be suitable for all membrane protein systems due to their low labelling efficiency, expression limitations, misfolding, or loss of function.~~

[Comment 6]

The authors should also state in the results section the exact identify of the employed construct. This information is in the methods section but also needs to be stated up front in the results.

Related to this, in the methods section the authors cite a 2018 Nature Chem Bio paper that employed the same protein and purification strategy. However, the receptor in the 2018 study also included an S110R stabilizing point mutation. If this was used in the current study it needs to be stated explicitly both in the results and methods sections. The 2018 Nature Chem Bio paper also states that the S110R mutation increases the affinities for antagonists. This would certainly impact the presented data, especially the data in Figure 4. Therefore this absolutely needs to be clarified in the text, otherwise no one will be able to reproduce these data if they are using a receptor without the S110R mutation.

[Reply]

Thank you for your useful comment. According to the suggestion, we revised the manuscript as follows, thank you.

In addition, in this study we used the wild-type M₂R fused with thermostabilized apocytochrome b562 (BRIL) at third intercellular loop (ICL3) that has previously been crystallized at resolution of 3.0 Å¹⁵ ~~Suno R. et al. (2018).~~ (lines 121-123 (revised manuscript))

[Comment 7]

The Figure 2 x-axis states “amide 1 percent population” but this does not make sense with the units. The left and right-most units are 0.0 and 2.0, which would then be 0% to 2%, which is not what I believe the authors intended.

[Reply]

We apologize for the confusion and corrected the word from “amide-I percent population” to “amide-I band ratio” for both main text and figures.

[Comment 8]

Regarding the data in Figure 4, it should be explained more clearly how these experiments are done. I’m assuming that at 30 minutes ligand is added to solution, but this is not explicitly described in the paper or made clear in the figure, which could be shown by indicating this time point with an arrow. Other important experimental details are also not described for these experiments, such as the flow rate, the amount of ligand added, when ligand was added.

[Reply]

We apologize for the lack of explanation and added the detail information of time point for ligand injection, buffer flow rate, and ligand concentration in Fig. 4 and its figure legend as follow, thank you.

Each ligand (1 mM concentration) is added at 30 min time point and is washed out at 90 min time point by exchanging to the buffer without ligand at a flow rate of 0.6 mL min⁻¹ through a flow cell, of which the temperature is maintained at 20 °C by circulating water. The arrows in the illustration mark the time points of ligand-In and ligand-out via buffer exchange, respectively. (lines 684-687 (revised manuscript))

[Comment 9]

Also related to Figure 4, the authors should report what effective off rates were calculated based on this data and it would be interesting to compare those number with any available literature data from surface plasmon resonance or other techniques, if it is publicly available (if not, then it would be acceptable to not include this comparison).

[Reply]

Thank you for suggestion. Taking account of the reviewer's comments, we have calculated all the ligand dissociation rate constants (k_{off}) from receptor by fitting curves with single exponential function in Fig. 4b (above) and the calculated k_{offs} for each ligand

ligand	k_{off} (min^{-1})
ACh	0.13 ± 0.02
MCh	0.18 ± 0.03
Carb	0.17 ± 0.02
Are	0.005 ± 0.001
Ixo	0.02 ± 0.001
Pilo	0.16 ± 0.05
McN	0.15 ± 0.02
Xano	0.01 ± 0.001
Atro	0.003 ± 0.0004
Sco	0.002 ± 0.0005
Ipra	N.D.*
NMS	0.01 ± 0.006
Tio	0.003 ± 0.0008

*Indicates no dissociation from M₂R at least in the experimental conditions used in this study.

were summarized in Table. 1 as follows. Other literature are NOT publicly available.

[Comment 10]

Regarding the model presented in Figure 5, unless assignments of the fine structure have been confirmed, this model is not presently supported by the data. In the 2019 J Phys Chem Lett paper, the authors tentatively assigned several lines to specific amino acids or groups of amino acids, such as the tyrosine lid. In the present work, the authors assume these assignments are correct, even though there were stated as being tentative assignments only in the 2019 paper. The spectra are complex and there are many structural changes that occur in GPCRs between antagonist-bound and agonist-bound conformations, including changes for the same amino acid types that have been tentatively assigned but in regions of the protein that are different than what the authors claim. Without more definitive assignment information, showing this model makes too many assumptions and therefore it should be removed.

[Reply]

Thank you for suggestion. Indeed, we have recently assigned the band at 1246 cm^{-1} as the mixed band of C-O stretch of ACh and possibly C-O stretch of tyrosine lid by using $2\text{-}^{13}\text{C}$ labelled ACh (Suzuki, K., Katayama, K., Sumii, Y., Nakagita, T., Suno, R., Tsujimoto, H., Iwata S., Kobayashi, T., Shibata, N. & Kandori, H. Vibrational analysis of acetylcholine binding to the M_2 receptor. *RSC Adv.* **11**, 12559-12567 (2021)). Based on the results, we proposed that the presence of the 1246 cm^{-1} band could be one of the indicators to distinguish ligand efficacy (response to comment 2). Hence, we decided to retain the model illustrated in Fig. 5, but we also added the summary of the three factors for examining the ligand efficacy found in this study in Fig. 5 as follow.

Replies to the Comment of Reviewer 3

We are thankful to you for your fully positive comments on our manuscript, which are very helpful to us in improving the manuscript.

We reply to your comments as follows.

Major comments

[Comment 1]

Page 4, second paragraph - please refer to the previous findings of Vogel et al. for rhodopsin here.

[Reply]

Thank you for your useful comment. According to the suggestion, we revised the manuscript and added the reference as follows, thank you.

(Line 73-85 (revised manuscript)) Fourier transform infrared (FTIR) spectroscopy has also been successfully applied to examine the structural and functional properties of the photoreceptive GPCR, rhodopsin. Light stimulus-induced difference FTIR spectroscopy combined with low temperature or specific pH values has unveiled various molecular events in the photoactivation processes upon light absorption. The sequential helix movements were monitored by amide-I band corresponding to mostly protein backbone amide carbonyl (C=O) stretching vibration^{24,25}, local changes in hydrogen bonding were deduced from characteristic C=O stretches of protonated carboxylic acid groups^{24,25}, and protein bound water molecules were detected by water O-H/O-D stretching vibrational changes²⁶ (Furutani Y. et al. (2003)). ~~However, both spectroscopy techniques require large quantities of pure protein, isotopic labelling and/or site directed mutations, which might not be suitable for all membrane protein systems due to their low labelling efficiency, expression limitations, misfolding, or loss of function.~~

24: Mahalingam, M., Martínez-Mayorga, K., Brown, M. & Vogel, R. Two protonation switches control rhodopsin activation in membranes. *Proc. Natl. Acad. Sci. U. S. A.* **105**, 17795-17800 (2008).

25: Zaitseva, E., Brown, M. F. & Vogel, R. Sequential rearrangement of interhelical networks upon rhodopsin activation in membranes: The Meta IIa conformational substate. *J. Am. Chem. Soc.* **132**, 4815-4821 (2010).

26: Furutani, Y., Shichida, Y. & Kandori, H. Structural changes of water molecules during

the photoactivation processes in bovine rhodopsin. *Biochemistry* **42**, 9619-9625 (2003).

[Comment 2]

Pag 5, second paragraph - with regard to the amide-I band please refer to Vogel et al. papers.

[Reply]

We appreciate the suggestion and added the reference as described above (Line 78, 79 (revised manuscript)).

[Comment 3]

Page 7, end of first paragraph - here it is essential to refer to the above-mentioned Vogel et al. papers, which show how the rhodopsin activation equilibrium is shifted upon conversion of retinal from an inverse agonist to an agonist.

[Reply]

We appreciate the suggestion and added the text/reference as follows, thank you.

(Line 169,170 (revised manuscript)) Thus, the spectral shift pattern of the amide-I band suggests that it is reflected in the conformational equilibrium between the inactive state and active states of M₂R as indicated from previous FTIR study of rhodopsin^{24,25}.

[Comment 4]

Page 10, top paragraph - please refer to above mentioned Vogel et al. papers.

[Reply]

We added a reference (Mahalingam et al. *Proc. Natl. Acad. Sci. USA.*, 2008, Zaitseva et al. *J. Am. Chem. Soc.*, 2010) at line 251. Thank you.

[Comment 5]

Page 11 beginning of discussion - please refer to Vogel et al.

[Reply]

We added a reference (Mahalingam et al. *Proc. Natl. Acad. Sci. USA.*, 2008, Zaitseva et al. *J. Am. Chem. Soc.*, 2010) at line 308. Thank you.

REVIEWERS' COMMENTS:

Reviewer #1 (Remarks to the Author):

The authors have addressed by points. There is one sentence on p3

"This is partly because of their low affinities to GPCR receptors resulting in conformational heterogeneity."

I do not agree and I perhaps was not clear. I was expecting a sentence like:

This is partly because efficacy of a ligand is thought to be reflected in changes to conformational equilibria, and thus the presence of multiple states.

Reviewer #2 (Remarks to the Author):

Katayama et al. have submitted a revised manuscript and corresponding response to the reviewers. After reading through the comments in the response to all three reviewers, I conclude that they have largely addressed the concerns that were raised during the first submission. I therefore support publication of the revised manuscript, provided that one minor additional revision is made.

In the SI Table 1, the stated standard errors in the means have higher precision (i.e. more significant digits) than the reported means, which should be corrected. I don't need to see the manuscript again to confirm that this has been addressed. I assume the authors will address this.

Replies to the Comment of Reviewer 1

We are thankful to you for your valuable comments and suggestions, which are very helpful to us in improving the manuscript.

We reply to your comments as follows.

[Comment 1]

"This is partly because of their low affinities to GPCR receptors resulting in conformational heterogeneity."

I do not agree and I perhaps was not clear. I was expecting a sentence like:

"This is partly because efficacy of a ligand is thought to be reflected in changes to conformational equilibria, and thus the presence of multiple states. "

[Reply]

Thank you for your useful comment. We revised the manuscript as follows.

P3 Line 65:

This is partly because efficacy of a ligand is thought to be reflected in changes to conformational equilibria, and thus the presence of multiple states. ~~This is partly because of their low affinities to GPCR receptors resulting in conformational heterogeneity.~~ In addition, these structural methods of X-ray crystallography and cryoEM analysis capture only a snapshot, low-energy conformation, which lack the conformational heterogeneity, therefore these methods cannot fully explain the mechanism of the efficacies.

Replies to the Comment of Reviewer 2

We are thankful to you for your critical reading of our manuscript. We reply to your comments as follows.

[Comment 1]

In the SI Table 1, the stated standard errors in the means have higher precision (i.e. more significant digits) than the reported means, which should be corrected.

[Reply]

We revised the SI Table 1 accordingly, thank you.

ligand	k_{off} (min ⁻¹)
ACh	0.13 ± 0.02
MCh	0.18 ± 0.03
Carb	0.17 ± 0.02
Are	0.005 ± 0.001
Ixo	0.020 ± 0.001
Pilo	0.16 ± 0.05
McN	0.15 ± 0.02
Xano	0.010 ± 0.001
Atro	0.003 ± 0.000
Sco	0.002 ± 0.000
Ipra	N.D.*
NMS	0.010 ± 0.006
Tio	0.003 ± 0.000

*Indicates no dissociation from M₂R at least in the experimental conditions used in this study.